# Stem Xylem Differences in Congeneric Lianas Between Forests Are Unrelated to Hydraulic Safety but Partly Explain Efficiency

**DOI:** 10.3390/plants14192951

**Published:** 2025-09-23

**Authors:** Caian S. Gerolamo, Anselmo Nogueira, Luciano Pereira, Steven Jansen, Elisangela X. Rocha, Veronica Angyalossy

**Affiliations:** 1Departamento de Ciências Biológicas, Universidade Regional do Cariri, Crato 63105-000, CE, Brazil; 2Centro de Ciências Naturais e Humanas (CCNH), Universidade Federal do ABC, São Bernardo do Campo 09606-070, SP, Brazil; a.nogueira@ufabc.edu.br; 3Institute of Botany, Ulm University, Albert-Einstein-Allee 11, D-89081 Ulm, Germany; biolpereira@gmail.com (L.P.); steven.jansen@uni-ulm.de (S.J.); 4Instituto Nacional de Pesquisas da Amazônia (INPA), Manaus 69011-970, AM, Brazil; elisangelarocha.xavier@gmail.com; 5Departamento de Botânica, Universidade de São Paulo, São Paulo 05508-090, SP, Brazil; vangyalossy@usp.br

**Keywords:** climbing plant, drought, vulnerability to embolism, P50, bordered pit structures, wood anatomy

## Abstract

Lianas are known for their distinctive vascular anatomy and remarkable hydraulic efficiency. Yet they exhibit considerable variation in hydraulic safety across and within forest types. This observation suggests different structure–functional strategies among lianas growing under contrasting levels of drought. Here, we compared xylem features at the cellular and intervessel pit levels and investigated their relationships with hydraulic safety and efficiency in five pairs of congeneric Bignonieae lianas from a seasonally dry forest (SDF) and a wet rainforest (RF). We hypothesize that rainforest lianas have xylem traits that maximize conductivity, while lianas from seasonally dry forests show greater woodiness and investment in storage tissues, and that xylem features at different levels drive the hydraulic safety and efficiency. The SDF liana species had a higher vessel density and grouping, and thinner fibers than rainforest lianas, but none of the features measured were related to hydraulic safety. Our results do not support that vessel or pit quantitative properties predict hydraulic safety in lianas. However, a higher hydraulic vessel diameter, total pit membrane area, and lower intervessel wall–lumen ratio were associated with high hydraulic efficiency, regardless of the forest type. These findings highlight the complexity of hydraulic structure–function relationships in lianas. While we found distinct xylem anatomical differences between species from contrasting forest types, only some traits were associated with hydraulic efficiency, and none predicted hydraulic safety, suggesting that other factors may be at play.

## 1. Introduction

By acting as ecological filters, environmental conditions influence which species persist in a given habitat, ultimately shaping both the functional and taxonomic composition of plant communities [1,2,3]. The variation in water availability in tropical forests was a key environmental feature that may determine contrasting anatomical variation and may explain plant distribution patterns [1,3,4]. Lianas have interesting anatomical features, such as wide vessels and typically high amounts of parenchyma, which provide a high efficiency in transporting water and very flexible stems, contributing also to their rapid growth [5,6,7,8]. Vessels, parenchyma, and fibers are the main cell types that compose xylem tissue. The xylem is part of the vascular system and performs multiple functions, including long-distance water transport, mechanical support, and storage [4,9,10]. Xylem anatomical traits offer insights into tissue structure and function, as well as into the ecological strategies of species [4,11,12]. Therefore, studying the ecology of lianas in forests with contrasting levels of water availability requires research on how and to what extent forest types shape xylem anatomical characters and the functional strategies associated with these features.

Many ecological studies of wood indicate that plants in moist environments tend to have wide vessels. In contrast, those in dry environments generally have smaller vessel diameters [4,13,14], a pattern influenced both by plant height and water conditions [15]. Indeed, the vessel diameter is a central feature in the apparent trade-off between hydraulic safety and efficiency [16,17,18]. The direct and exponential relationship of vessel diameter with water conductivity, according to Hagen–Poiseuille’s law, is well established and explains the gain in conduction efficiency [19,20,21]. On the other hand, the relationship between vessel diameter and drought-induced embolism resistance, i.e., hydraulic safety, remains unclear and is mechanistically not well understood. Vessel diameter is likely indirectly related to embolism resistance [22]. Intervessel pit membrane thickness is known to influence embolism resistance by affecting the number of pore constrictions in pore pathways across a pit membrane [8,23,24,25], while the total amount of intervessel pits per area may or may not play a role in embolism resistance [25,26].

A functional link between vessel diameter and drought-induced embolism resistance was proposed in the “pit area hypothesis” [13,17,18]. This hypothesis proposes that wide vessels have larger total intervessel pit membrane areas per vessel wall area (A_P_) than narrow vessels. This higher A_P_ may, in turn, lead to lower embolism resistance. However, the available evidence supporting this relationship remains limited [17,18]. Whether or not variation in A_P_ affects embolism resistance depends on pit membrane thickness. An intermediate to thick pit membrane may at least partly uncouple hydraulic safety from efficiency, both at the individual pit membrane level and vessel level [24,25,26]. The pit area hypothesis may apply to plants that have an intermediate pit membrane thickness, as could be the case with lianas [8]. In a wet habitat, lianas are expected to have a high vessel diameter and Ap to minimize hydraulic resistance, but also embolism resistance.

A second structural–functional relationship described in various studies on ecological wood anatomy is that plants in environments with frequent droughts should have a higher vessel density and grouping of vessels, in addition to thick vessel walls and fibers, thereby conferring greater woodiness [4,27,28,29]. A higher grouping of vessels, as seen in a transverse wood section, would increase the network of intervessel connections if vessel diameter and length were similar. Traditionally, a higher vessel grouping is interpreted to provide alternative pathways for bypassing water if a vessel is embolized [28,29,30]. Although some studies provide experimental [31,32] and theoretical [20,33] evidence that plants with a high vessel grouping are more resistant to drought-induced embolism, this relationship remains poorly understood. Furthermore, vessel wall thickness, the amount and chemical composition of fiber walls, and the parenchyma matrix surrounding the vessel network can contribute to embolism resistance [34,35,36,37] and with mechanical support [4,9,11]. We know so far that lianas of seasonal dry forests follow some of these patterns by having a higher vessel density [8] and a lower fiber tissue fraction than lianas of rainforests [38]. Clearly, more work is needed in testing how vessel density and vessel grouping may relate to embolism resistance.

In this context, this study aims to compare xylem structure at multiple scales, from cells to intervessel pits, and to examine how these structural traits relate to hydraulic safety and efficiency in lianas growing under contrasting levels of water availability. For this, we addressed two key questions: (i) Are there wood anatomical differences in lianas from forest types with contrasting levels of water availability? (ii) Which xylem features of lianas predict safety and hydraulic efficiency in tropical forests? To answer these questions, we chose five pairs of congeneric liana species of the tribe Bignonieae distributed in two distinct Neotropical forest types: the wet Amazon rainforest (~2 dry months per year) and Atlantic seasonally dry forest (~6 dry months per year). Using congeneric species ensures phylogenetic independence and minimizes confounding effects in comparative analyses [39]. We propose two hypotheses: (H1) Rainforest lianas have xylem traits that maximize conductive capacity, whereas lianas from seasonally dry forests show traits reflecting greater woodiness and investment in storage tissues; (H2) We expect that xylem features, from the cellular level to the intervessel pits, determine the hydraulic safety and efficiency of liana species, enabling them to cope with local water conditions.

## 2. Results

### 2.1. Qualitative Characterization of Lianescent Xylem

The young branches of all Bignonieae lianas have a cambial variation with xylem interrupted by four phloem wedges regardless of forest type (Figure 1A,B,E,I,M,Q,U, Figure 2A,E,I,M, and Appendix A). The bark anatomy was similar between liana species of rainforest and seasonally dry forest, and was formed by the periderm, cortex, pericyclic fibers, and secondary phloem (Appendix A).

The secondary xylem of all species, regardless of forest type, had vessel dimorphism, i.e., large vessels and many very small diameter vessels (Figure 1C,G,J,K,R,S,X, Figure 2B,F,G,J,N,K,O and Appendix A). In the innermost portion of the secondary xylem, lianas have vessels with a small diameter (<50 µm), which are generally arranged in radial chains, and a high proportion of fibers (Figure 1B,F,R and Figure 2A,B,M,N). In the outermost portion, close to the cambium, the xylem was marked by large vessels grouped in different arrangements and a higher vessel density than the innermost portion. This anatomical configuration of the innermost and outermost xylem portions occurred in all lianas regardless of forest type. Intervessel pits were bordered and alternate in all lianas (Figure 1D,H,L,P,T,Z and Figure 2D,H,L,P). Axial parenchyma was predominantly paratracheal, scarce in liana species of the rainforest (Figure 1C,K,S and Figure 2C,K), and predominantly paratracheal vasicentric in liana species of the seasonally dry forest, which eventually form short confluences (Figure 1G,O,X and Figure 2G,O). In *Fridericia triplinervia*, *Tynanthus panurensis*, and *Tynanthus fasciculatus*, marginal parenchyma in differentiation near the vascular cambium begins to form in some individuals (Figure 2F,N). Fibers are septate with simple pits (Appendix A). Rays include square, upright, and procumbent cells mixed throughout the ray (Appendix A), and are one to three cells wide in all liana species (Appendix A). We found perforated ray cells in *Fridericia triplinervia* and *Tynanthus panurensis* (Appendix A). The ray–vessel pits are similar to the intervessel pits (Appendix A). We noticed many starch granules and crystals in the fibers and axial and ray parenchyma in all species, regardless of the forest type (Appendix A). We did not find tracheids in any of the sampled liana species (Appendix A).

### 2.2. Coordination of Quantitative Xylem Features of the Lianas Studied

Average values (±SD) of xylem features for each liana species of the rainforest and seasonally dry forest are shown in Table 1 and Appendix A.

**Table 1 plants-14-02951-t001:** Average (±SD) of xylem features at the cell and intervessel pit level of five pairs of congeneric liana species of Bignonieae occurring in two contrasting forests (rainforest and seasonally dry forest). For more details, see Table 2.

	Rainforest					Seasonal Dry Forest				
	*Adenocalymma validum*	*Anemopaegma robustum*	*Bignonia aequinoctiales*	*Fridericia triplinervia*	*Tynanthus panurensis*	*Adenocalymma bracteatum*	*Anemopaegma chamberlynii*	*Bignonia campanulata*	*Fridericia triplinervia*	*Tynanthus fasciculatus*
Apex distance (m)	1.02 ± 0.56	1.27 ± 0.57	0.93 ± 0.17	1.30 ± 0.66	0.96 ± 0.12	1.35 ± 0.67	1.42 ± 0.38	1.46 ± 0.58	1.23 ± 0.42	1.18 ± 0.48
Stem diameter (mm)	6.33 ± 0.45	6.83 ± 0.79	6.92 ± 0.63	3.83 ± 0.65	7.81 ± 1.95	3.23 ± 0.58	4.18 ± 1.21	3.21 ± 0.43	4.94 ± 1.42	3.73 ± 0.60
Vessel area percentage	0.32 ± 0.05	0.22 ± 0.05	0.23 ± 0.09	0.34 ± 0.11	0.19 ± 0.07	0.33 ± 0.07	0.31 ± 0.06	0.37 ± 0.10	0.29 ± 0.14	0.28 ± 0.05
Fiber area percentage	0.43 ± 0.05	0.51 ± 0.05	0.49 ± 0.10	0.43 ± 0.16	0.52 ± 0.06	0.37 ± 0.07	0.44 ± 0.08	0.38 ± 0.10	0.44 ± 0.12	0.43 ± 0.08
Parenchyma area percentage	0.25 ± 0.03	0.27 ± 0.05	0.26 ± 0.03	0.23 ± 0.05	0.30 ± 0.02	0.27 ± 0.02	0.22 ± 0.05	0.24 ± 0.02	0.27 ± 0.05	0.27 ± 0.06
Vessel diameter (μm)	31.4 ± 5.31	36.1 ± 7.64	33.2 ± 12.1	36.9 ± 8.01	30.2 ± 5.44	30.3 ± 5.46	29.8 ± 4.90	32.6 ± 7.15	40.0 ± 9.14	33.8 ± 4.65
Maximum vessel diameter (μm)	117 ± 29.3	109 ± 22.5	101 ± 36.5	107 ± 43.6	90.5 ± 9.10	79.7 ± 5.15	87.4 ± 12.5	69.3 ± 13.9	109 ± 33.3	91.3 ± 15.3
Minimum vessel diameter (μm)	9.92 ± 1.93	10.2 ± 1.71	11.2 ± 2.89	9.01 ± 1.18	10.6 ± 0.90	9.70 ± 1.32	10.9 ± 1.38	9.48 ± 1.59	12.3 ± 2.35	10.3 ± 1.63
Vessel density (n mm^−2^)	88.3 ± 17.3	85.0 ± 22.3	69.4 ± 26.0	123 ± 23.3	89.5 ± 17.0	160 ± 63.0	164 ± 53.3	197 ± 60.3	101 ± 20.7	100 ± 23.3
Hydraulic vessel diameter (µm)	61.8 ± 13.0	57.4 ± 10.7	55.4 ± 19.6	56.4 ± 15.5	48.6 ± 83.4	44.9 ± 24.6	50.5 ± 68.9	42.3 ± 10.2	59.4 ± 13.1	51.0 ± 66.8
Vessel grouping index	1.90 ± 0.51	1.79 ± 0.37	1.88 ± 0.34	1.89 ± 0.40	2.11 ± 0.78	2.14 ± 0.58	3.29 ± 0.73	2.13 ± 0.52	1.87 ± 0.22	2.16 ± 0.71
Intervessel wall–lumen ratio	0.003 ± 0.002	0.003 ± 0.001	0.007 ± 0.005	0.003 ± 0.001	0.006 ± 0.002	0.006 ± 0.001	0.004 ± 0.001	0.012 ± 0.003	0.006 ± 0.003	0.004 ± 0.002
Vessel wall thickness (μm)	6.59 ± 0.93	6.38 ± 1.26	7.14 ± 1.86	6.14 ± 1.95	6.97 ± 1.67	6.32 ± 1.03	6.06 ± 1.07	7.48 ± 0.99	8.01 ± 0.83	5.66 ± 0.56
Fiber wall thickness (μm)	4.83 ± 0.28	5.21 ± 0.79	4.47 ± 0.42	4.36 ± 0.49	4.78 ± 0.89	3.02 ± 0.30	2.89 ± 0.36	2.93 ± 0.18	3.72 ± 0.51	2.63 ± 0.43
Ray width (μm)	11.5 ± 1.68	16.6 ± 2.79	15.1 ± 3.34	9.54 ± 1.70	13.7 ± 4.36	10.4 ± 1.71	12.8 ± 1.36	11.6 ± 1.81	13.1 ± 3.11	8.67 ± 0.80
Ray height (μm)	423.3 ± 183.8	719.9 ± 274.2	817.9 ± 194.7	429.1 ± 142.2	518.3 ± 192.8	423.3 ± 183.8	776.1 ± 228.0	513.7 ± 220.2	484.8 ± 111.5	377.4 ± 126.8
Pit membrane surface area (μm^2^)	24.0 ± 4.42	18.8 ± 3.93	24.7 ± 8.94	26.6 ± 8.75	16.6 ± 1.87	35.2 ± 9.87	28.6 ± 3.16	22.8 ± 4.85	30.6 ± 7.17	19.7 ± 2.54
Pit aperture (μm^2^)	1.34 ± 0.18	1.28 ± 0.29	1.32 ± 0.41	1.16 ± 0.54	0.83 ± 0.10	1.82 ± 0.42	1.70 ± 0.21	1.32 ± 0.57	1.40 ± 0.41	0.96 ± 0.25
Intervessel contact fraction	0.08 ± 0.01	0.06 ± 0.01	0.06 ± 0.01	0.09 ± 0.01	0.12 ± 0.02	0.12 ± 0.04	0.08 ± 0.01	0.10 ± 0.01	0.11 ± 0.02	0.10 ± 0.02
Pitfield fraction	0.70 ± 0.03	0.70 ± 0.05	0.71 ± 0.07	0.68 ± 0.03	0.64 ± 0.02	0.69 ± 0.04	0.70 ± 0.03	0.67 ± 0.04	0.64 ± 0.05	0.58 ± 0.05
Pit fraction	0.06 ± 0.01	0.04 ± 0.00	0.04 ± 0.00	0.06 ± 0.00	0.07 ± 0.01	0.08 ± 0.03	0.06 ± 0.00	0.06 ± 0.01	0.07 ± 0.01	0.06 ± 0.01
Pit density (n° 200 μm^2^)	3.33 ± 0.51	4.5 ± 1.04	4 ± 1.55	2.71 ± 0.48	5.16 ± 0.98	1.41 ± 0.49	2 ± 0	3.08 ± 1.02	1.75 ± 1.17	2.68 ± 0.70
Total pit membrane area (10^−6^ m^2^)	1.18 ± 0.45	1.23 ± 0.61	0.90 ± 0.72	1.87 ± 0.53	1.64 ± 0.60	2.13 ± 1.29	1.36 ± 0.35	1.41 ± 0.72	2.00 ± 0.92	1.46 ± 0.49

**Table 2 plants-14-02951-t002:** List of xylem and hydraulic features, abbreviations, and descriptions.

Abbreviation	Feature and Description	Unit
Vessel tissue fraction	Percentage of vessel lumen area	%
Fiber tissue fraction	Percentage of fiber area	%
Parenchyma tissue fraction	Percentage of parenchyma area (including ray and axial parenchyma)	%
D_v_	Vessel diameter (maximum and minimum)	µm
D_h_	Hydraulic diameter of vessel [=(${\sum D^{4}/N)}^{1/4}$]	µm
VD	Vessel density	N mm^−2^
GI	Vessel grouping index: total vessels divided by the total of vessel groups with true intervessel walls; a solitary vessel counts as one vessel group	-
(T_vw_/D_max_)^2^	Intervessel wall–lumen ratio: double intervessel wall thickness (T_vw_) divided by the maximum diameter of the vessel (D_max_) squared	-
L_V_	Mean vessel length	m
S_pit_	Intervessel pit membrane surface area	µm^2^
D_pit_	The number of intervessel pits per 200 µm^2^	N 200 µm^2^
F_C_	Intervessel contact fraction	%
F_PF_	Pitfield fraction, i.e., the ratio of the pit membrane area to the intervessel wall area	%
F_pit_	Intervessel pit fraction of an intervessel wall (=F_C_·F_PF_)	%
A_P_	Total intervessel pit membrane area for a vessel with average diameter and length (=π·D_V_·L_V_·F_pit_)	m^2^
T_F_	Fiber wall thickness	µm
P50	Water potential at which 50% of the maximum amount of gas is discharged	MPa
Ks	Theoretical specific hydraulic conductivity	Kg m^−1^ MPa^−1^ s^−1^

The principal component analysis (PCA) evidenced a covariation among xylem features and defined two principal axes describing this covariation. Both axes explain 50% of the data covariance (Figure 3). The first axis (PC1-27.8%) was positively related to vessel density and vessel grouping index and negatively associated with fiber thickness and hydraulic vessel diameter (Figure 3 and Appendix A). Vessel density, vessel grouping index, and fiber thickness contributed significantly to PC1 (loadings > ±0.6; *p* < 0.05), whereas hydraulic vessel diameter showed a marginally significant contribution (loading = 0.56; *p* = 0.06; Appendix A). The second axis (PC2-21.7%) was positively associated with the total pit membrane area, hydraulic vessel diameter, and pit membrane surface area, and negatively associated with the intervessel wall–lumen ratio [(T_vw_/D_max_)^2^] (Figure 3; Appendix A). Total pit membrane area, hydraulic vessel diameter, and [(T_vw_/D_max_)^2^] contributed significantly to PC2 (loadings > ±0.6; *p* < 0.05), whereas pit membrane surface area showed a marginally significant contribution (loading = 0.56; *p* = 0.06; Appendix A). In general, we found an effect of forest type, genus, and interaction between forest type and genus for the first PCA axis (Figure 4A, Appendix A). Three congeneric pairs of Bignonieae lianas from the seasonal forest had higher PC1 values than the congeneric liana species of the rainforest (Appendix A). In contrast, *Fridericia* and *Tynanthus* congeneric pairs had similar PC1 values between the two forest types (Figure 4; Appendix A). The relationships between PC1 and forest type (Figure 4A) show a coordinated shift in xylem feature profiles along PC1, where liana species of the drier seasonal forest clustered on one side (right) of the ordination space, driven by high vessel density and vessel grouping index. The liana species of the rainforest on the left side were driven by higher fiber thickness (Figure 4A). We found an effect of genus for PC2 (Figure 4B; Appendix A), but there was no effect of the forest type.

### 2.3. Testing of a Relationship Between Hydraulic and Xylem Anatomy

We did not find a relationship between the xylem features, summarized by PC1 and PC2, and hydraulic safety (P50) (*p* > 0.05, marginal R^2^ = 0.08, Figure 5, Appendix A). On the other hand, theoretical specific conductivity (Ks) was related to PC2, in which higher values of Ks were associated with more positive values of PC2 (b_st_ = 0.68 ±0.06, t = 10.35, *p* < 0.001, marginal R^2^ = 0.65, N = 57), which represents stems with a combination of higher total pit membrane area and hydraulic vessel diameter concomitant with a lower intervessel wall–lumen ratio (Figure 5D, Appendix A).

## 3. Discussion

We characterized the xylem structure at the cell and intervessel pit level and associated it with the hydraulic safety and efficiency of five pairs of congeneric liana species of the tribe Bignonieae that occur in a seasonally dry forest and rainforest. Our results partially support our hypotheses by showing that liana species from seasonally dry forests have differences in their xylem anatomy, especially with respect to vessel density and grouping, and fiber wall thickness (Figure 4A). However, none of the features measured directly determined hydraulic safety in our study group (Figure 5A,B). On the other hand, hydraulic vessel diameter, total pit membrane area, and intervessel wall–lumen ratio did not change with the type of forest, and were the features related to hydraulic efficiency. Our results highlight the existence of a coordinated set of features of the vessel and intervessel pits in the liana species of tropical forests to regulate hydraulic efficiency (Figure 5D).

### 3.1. Lianas from Tropical Forests with Different Water Regimes Have Divergent Xylem Features

Bignonieae lianas from a seasonally dry forest have xylem features that are more similar to each other and different from congeneric liana species of a rainforest. An exception to this was the genus *Fridericia* and *Tynanthus*. Our analyses showed a higher vessel density and vessel grouping index, associated with a lower wall thickness of fibers in liana species from a seasonally dry forest compared with liana species growing in a rainforest. The high vessel density and vessel grouping index in a seasonally dry forest are consistent with patterns reported for other liana species from neotropical forests that experience contrasting water availability [8]. In general, a high vessel density and vessel grouping are common features of plants in drier environments, reflecting greater woodiness [3,28,40,41]. However, the low fiber tissue fraction and relatively thin fiber walls in liana species of seasonally dry forests indicate an opposite pattern to what has been found in trees and shrubs of dry environments [38,42,43]. Relative thin fiber walls and high vessel density in liana stems can increase mechanical flexibility, favoring climbing to the forest canopy and reducing the chances of stem damage [11,44,45].

We found that the vasicentric axial parenchyma with the presence of starch was another outstanding feature of young branches of liana species from the seasonally dry forest compared to congeneric liana species of the rainforest. Liana species from the rainforest also had starch, mainly in wide rays of young branches, but the axial parenchyma was less pronounced and scarcely paratracheal. Although our findings showed that the type of axial parenchyma is different in young branches of Bignonieae lianas from seasonally dry forest and rainforest, the proportion of total parenchyma (axial and radial) in xylem was similar, as found for other lianas of neotropical forests [38]. The presence of high and wide radial parenchyma and vasicentric axial parenchyma might facilitate long-distance water transport, xylem hydraulic capacitance, carbohydrate storage (e.g., starch), and the repair and prevention of vascular tissue disruption after injury [35,44,46,47,48].

We found no difference with the forest type in the hydraulic vessel diameter, intervessel wall–lumen ratio, and total pit membrane area, summarized by PC2, but differences among the genera. Similarly, another study found no difference in mean nor maximum vessel diameter between dry forest lianas and rainforest lianas, which implies a similar hydraulic conductive capacity [8,10]. Also, liana vessels of both forests have a similar total pit membrane area. Data quantifying these features of intervessel pit structure in lianas are scarce, and further studies exploring these features and the ultrastructure of bordered pits may help us understand whether there are recurrent anatomical patterns in lianas from different environments. Therefore, our results indicate that, while certain anatomical features of vessels, fibers, and axial parenchyma were different in Bignonieae lianas across forest types, other variations could reflect taxonomic differences and did not reflect the type of forest. The vessel and intervessel pit features play a key hydraulic functional role in the safety and efficiency of water transport [8,22,31,32], as discussed below.

### 3.2. Linking Anatomical Features with Hydraulic Safety and Efficiency

We found no relationship between the xylem features measured, summarized by PC1 and PC2, and hydraulic safety in Bignonieae lianas. This result does not support the vessel diameter and hydraulic safety trade-off found for many trees and shrubs, which predicts that plants with higher hydraulic vessel diameters are less resistant to drought-induced embolism [16,17,32]. However, experimental data show a weak or indirect correlation between vessel diameter and hydraulic safety [8,23,31,49]. We found no evidence that vessel density and grouping contribute to increased resistance to drought-induced embolism, as speculated by [28]. Similarly, our results do not support the pit area hypothesis [17,18]. The absence of a relationship between xylem structure and hydraulic safety could indicate that multiple structural strategies drive hydraulic safety. For example, the study [11] demonstrated that diverse anatomical combinations in liana stems can yield functionally equivalent designs for stem flexibility, in which similar functional outcomes arise from different anatomical pathways. Alternatively, the lack of support for the pit area hypothesis, as well as for vessel dimension and density, could reflect the influence of other anatomical traits more directly related to hydraulic safety, such as pit membrane thickness and gas–liquid–surfactant interactions [22,24,49,50]. Contrary to expectations, lianas are as hydraulically safe as trees [8,10], as having different sets of anatomical traits that ensure high hydraulic safety can contribute to lianas’ survival, also in drought-prone ecosystems.

Our results show that Bignonieae lianas with higher hydraulic vessel diameter and total intervessel pit membrane area associated with lower intervessel wall–lumen ratio increase hydraulic efficiency. The positive relationship between vessel diameter and hydraulic efficiency is well established and supported by previous studies with trees, shrubs, and lianas [6,51,52]. The Hagen–Poiseuille law for ideal capillaries determines that wide conduits result in an exponential increase in hydraulic conductivity in plants [21,28,52]. Lianas typically have large vessel diameters that can be associated with many small-diameter vessels, i.e., vessel dimorphism [21,53,54], as also found in our study. The many small-diameter vessels contribute very little to the total conductive efficiency but may be important to assist radial and tangential water transport [27,55]. In fact, narrow vessels could be numerous enough to maintain significant ‘back-up’ flow in case an adjacent wide vessel is embolized, similar to the role of vasicentric tracheids [56]. In addition, we found a positive relationship between the total intervessel pit membrane area and the increase in hydraulic efficiency in the Bignonieae lianas studied. The best explanation for this relationship found is that an increase in the total intervessel pit membrane area associated with thinner pit membranes reduces the resistance to water flow in plants, increasing transport efficiency [4,24,57,58]. On the other hand, the higher intervessel wall–lumen ratio reduces conductive efficiency in Bignonieae lianas studied due to increased resistance to water flow, as found for trees and shrubs [59,60,61]. Vessel wall thickness is positively related to pit membrane thickness, potentially reducing hydraulic efficiency compared with thin-walled vessels with thinner pit membranes [31,49]. Therefore, the vessel diameter associated with the intervessel pit membrane area and the thickness of the vessel wall was the main driver of hydraulic efficiency in Bignonieae lianas.

## 4. Conclusions

In conclusion, our study revealed that Bignonieae liana species from a seasonally dry forest exhibit higher vessel density and grouping, but lower fiber wall thickness, and a different type of axial parenchyma compared to their congeners from a rainforest. In addition, we found a coordinated set of vessels and intervessel pit features that may regulate hydraulic efficiency, regardless of forest type. However, none of the anatomical features measured predicted hydraulic safety, indicating that other structural factors may be involved. These findings highlight the complexity of hydraulic structure–function relationships in lianas. Our results expand the understanding of how and which xylem features affect the hydraulic properties of lianas in different environments, contributing to mechanistic models of plant hydraulics, including the lianas, which are abundant elements of tropical forests.

## 5. Materials and Methods

### 5.1. Study Site and Climate

The study survey was carried out at Ducke Reserve, a 10,000 ha terra-firme rainforest (RF) located in the central region of the Amazon basin, Manaus, Brazil (2°57′42″ S, 59°55′40″ W; [62]), and at the Santa Genebra Reserve, a 250 ha seasonally dry semideciduous forest (SDF) located in the Atlantic Forest domain in São Paulo, Brazil (22°49′20″ S, 47°06′40″ W; [63]).

The Ducke Reserve experiences a tropical climate ‘Am’ according to the Koppen–Geiger climate classification, with dry and rainy seasons ruled by monsoons [62]. The average annual temperature was 26 ± SD 10 °C, the humidity was around 84%, and the precipitation was 2572 ± 351 mm, with a rainy season from October to June, and a dry season from July to September [64,65]. The vegetation of the Ducke Reserve was an old-growth perennial rainforest, with emerging trees reaching 45 m, a closed canopy of 30-37 m, and a high diversity of tree and liana species [66,67,68]. The topography was heterogeneous, with soils forming a continuum of clayey oxisols in plateau portions, with increasing sand on the slopes until they become pure sand at the bottom of the valleys, forming small floodplains [69,70]. Elevations range from 40 to 140 m above sea level [67].

Santa Genebra Reserve experiences the tropical climate ‘Cwa’ according to the Koeppen–Geiger climate classification [63]. The average annual temperature was 22 ± 1 °C, the humidity was around 63%, and precipitation was 1360 ± 280 mm, with a dry and cold season from April to September, and a hot and rainy season from October to March, with September being one of the driest months of the year with precipitation of <50 mm month^−1^ [63]. The vegetation of the Santa Genebra Reserve was a seasonally dry semideciduous forest, occupying 85% of the reserve, with an almost continuous canopy at about 15 m in height, with emerging trees reaching 30 m and a high diversity of tree and liana species [63]. The topography is homogeneous, with elevations ranging from 580 to 610 m above sea level, and soils are typically dystrophic red argisol [71].

### 5.2. Species Collection

We selected ten Bignonieae liana species composed of five pairs of abundant congeneric species of lianas in each forest type (Appendix B). These species were chosen for being congeneric species pairs across forest types. Congeneric species pairs were utilized to evaluate whether drought-related traits associated with habitat were consistently present across different phylogenetic lineages. The genus *Fridericia* was an exception due to the presence of a species complex within *F. triplinervia*. We treated the two geographically separated and morphologically distinct populations of this species as congeneric pairs for subsequent analyses comparing both forest types. In our sampling, we included six to eight mature individuals per species. The specimens selected reached the forest canopy with a stem diameter > 1 cm at 1.30 m from the rooting point. Professional tree climbers sampled apical branches (ca. 2 m in length) for each specimen, and anatomical and hydraulic features were measured for one branch per individual. To avoid clonal plants, each liana was located more than 10 m from each other without any stem connection with other lianas [72].

### 5.3. Anatomical Measurements and Analyses

To perform the anatomical measurements and analyses, we sectioned samples from the base of the stem about 1 m from the apex using the same branches that were used for the hydraulic features. This procedure was used to establish a standard for the distance from the apex, which is necessary to compare the vessel diameter among other xylem features, irrespective of total plant height [73,74]. Each stem sample was fixed in FAA (formaldehyde, acetic acid, and 50% ethanol, [75]) for one day and then stored in 70% ethanol. The samples, including the bark (all tissues outside the vascular cambium) and secondary xylem, were embedded in polyethylene glycol 1500 (PEG-1500) [76]. Transverse, longitudinal, radial, and tangential sections of each sample were made with a sliding microtome [77]. Sections were stained in 1% Astra blue and 1% [78] modified by [79], and permanent slides were mounted for anatomical analyses. The wood description was based on the International Association of Wood Anatomists (IAWA) list of microscopic features of hardwood identification [80]. We applied macerations to investigate the presence of tracheids, i.e., imperforate tracheary elements with distinctly bordered pits [27]. Wood splinters were macerated in a 1:1 solution of glacial acetic acid and hydrogen peroxide at 60 °C for 48 h [81]. Dissociated cells were dyed with 1% safranin and mounted on temporary slides. Each histological slide was photographed with a photomicroscope (Leica DML-DFC 310FX), and the images were analyzed using ImageJ version 1.53a software (National Institutes of Health, Bethesda, MD, USA; http://rsb.info.nih.gov/ij/, accessed on 11 February 2022). For each stem histological slide representing an individual, the xylem was analyzed, and we estimated the vessel diameter (maximum and minimum), hydraulic vessel diameter, vessel density, vessel grouping index, intervessel wall–lumen ratio (double intervessel wall thickness divided by the maximum diameter of the vessel squared), intervessel contact fraction, and fiber wall thickness using the procedures established by [82]. We estimated the percentage of the area of vessels, fibers, and parenchyma, width, and height of rays using the procedures established by [6], see Table 2 for more details.

We used a scanning electron microscope (SEM) to characterize the intervessel pit dimension as pit density, pit aperture area, pit membrane area, and pitfield fraction. We followed the procedures established by [82] for measurement, and vessel–parenchyma pits were not included in these measurements. In each wood sample, we cut a piece of xylem in the tangential plane close to the vascular cambium with ca. 0.5 and 3 mm of thickness and width, respectively. These small wood samples were dehydrated in 90% ethyl alcohol (10 min), 95% alcohol (15 min), absolute alcohol (twice for 10 min), and dried with a critical point drier (model CPD 030 Balzers, Liechtenstein, Austria). The small wood samples were fixed to aluminum stubs with an electron-conductive carbon sticker and coated with gold using a sputter coater (SCD 050 Balzers, Liechtenstein, Austria) for 200 s. Small wood samples fixed to stubs were then observed with a field emission SEM at a voltage of 2–10 kV (Sigma VP Carl Zeiss, England), and images were taken with a digital camera (ADDA da Olympus). To estimate the pit density, pit aperture area, pit membrane area, and pitfield fraction, we used 3–4 vessels per sample, measuring about 25 intervessel pits per vessel. All analyses and anatomical measurements were accomplished in the interwedge regions, i.e., portions of the stem that have regular secondary growth in Bignonieae lianas. All anatomical measurements were conducted with ImageJ version 1.53a software.

Considering that the maximum vessel length was a significant predictor of mean vessel length in lianas [83], the mean vessel length of each Bignonieae liana was estimated based on the maximum vessel length measured using an air-injection method [56] and the linear model derived by [83]. The total surface area occupied by intervessel pit membranes (A_P_) was calculated following [17]: $A_{P}=\pi D_{V}L_{V}F_{PF}F_{C}$, where D_V_ was the mean vessel diameter, L_V_ was the mean vessel length, F_PF_ was the intervessel contact fraction, and F_C_ was the pitfield fraction. A complete list of hydraulic and xylem features with their abbreviations and descriptions is given in Table 2.

### 5.4. Hydraulic Safety

The hydraulic safety, i.e., embolism resistance, measurements for each plant were retrieved from the data used in [10]. Briefly, after collecting the branches from the liana canopy, each branch was sprayed with water, placed inside dark plastic bags with moistened paper towels, and the cut ends were immersed in a beaker with water to prevent desiccation. In the laboratory, on average, 35 min after the excision, we measured the xylem water potential of leaves over time to construct the vulnerability curves for each branch. The hydraulic vulnerability curves were constructed using the manual pneumatic method [84,85,86]. With this method, the relative amount of embolism in intact conduits by gas extraction was estimated from the increase in the volume of air inside the branch (xylem) caused by the formation of embolism as the branch dehydrates. Branches were bench dehydrated, and the xylem water potential (Ψ) was measured using a pressure chamber (PMS 1000; PMS Instruments Co., Albany, OR, USA; [87]). We pooled the data for each branch and fitted a sigmoid curve to the data, relating the percentage of gas discharged (PAD) to the xylem water potential. In the sigmoid curve, P50 and slope (*b*) are the fitted parameters in the model: PAD = 100/{1 + exp [*b* (Ψ − P50)]} [88]. We used the xylem water potential at which 50% of the maximum gas amount was discharged (P50) as an indicator of xylem resistance to hydraulic failure; for more details, see [10].

### 5.5. Hydraulic Efficiency

The hydraulic efficiency measurements for each plant were retrieved from the data used in [10], in which theoretical specific hydraulic conductivity (Ks) was used to describe water transport efficiency according to vessel diameter and vessel density [52,89]. To estimate theoretical specific hydraulic conductivity, we used at least 100 vessels and four areas of 1 mm^2^ to estimate the hydraulic vessel diameter [D_h_ = ($\sum {D^{4}/N)}^{1/4}$] and the vessel density (VD) of each individual, respectively. Using D_h_, VD, density of water (pw) at 20 °C (pw = 998.2 kg m^−3^), and the water viscosity (***η***) at 20 °C (***η*** = 1.002 × 10^−3^ Pa s^−1^), we calculated the theoretical specific hydraulic conductivity [Ks = $\left(\frac{\pi \cdot pw}{128\eta }\right)\cdot VD\cdot D_{h}^{4}$], following Hagen–Poiseuille’s law [52,89].

### 5.6. Data Analysis

To evaluate our first and second question, we first carried out a principal component analysis (PCA) with a set of nine xylem features that have been linked to hydraulic efficiency and safety (hydraulic diameter vessel, vessel grouping index, vessel density, intervessel wall–lumen ratio, total pit membrane area, pit membrane surface area, fiber wall thickness, fiber tissue fraction, and parenchyma tissue fraction). Before running the PCA, the quantitative variables describing the xylem structure were standardized by subtracting the mean and dividing by the standard deviation. For the PCA, individuals with missing trait values (n = 12) were excluded, resulting in a final dataset of 57 samples out of the 69 collected. The PCA summarized a set of correlated variables, and we selected the first two axes to represent the variation in the xylem structure based on randomization tests proposed by [90] (Appendix A). The randomization tests consisted of constructing 10,000 matrices from the randomized features to obtain 10,000 randomized eigenvalues to compare with the original eigenvalue obtained from the original xylem features [90]. In this case, the randomized matrices had no covariation among features, and the randomized eigenvalues had very low values. The selection of the xylem features that most contributed to the variation in the principal axes was also performed using a new set of randomization tests for each principal axis (Appendix A, [90]). Significant PCA eigenvalues and loadings, determined by the randomization test, are presented in Appendix A.

To evaluate whether there was an anatomical difference between climatically different forests, we assessed the effect of forest type and genus (representing the different lineages of lianas within the tribe Bignonieae) on the first two PCA axes (PC1 and PC2). For each PCA axis used as a response variable, we constructed a linear mixed model (LMM) with a Gaussian error distribution, including the forest type and genus as fixed factors. We also included in each model the species nested within a genus as the random term on the intercept to account for the non-independence of individuals nested in species and species nested by genus. When there was no interaction between factors in the models, a new model was built, only including the additive effect of each factor (forest type and genus). We expected that liana species from the seasonally dry forest compared to the liana rainforest had a lower vessel diameter, higher density and grouping of vessels, thicker vessel and fiber walls, higher proportion of fiber, and lower total intervessel pit area, which are features related to dry environments. Data on vessel density, vessel grouping index, total pit membrane area, and pit surface area were log-transformed before analysis to achieve normality.

To evaluate our second question about the relationship between xylem structure and hydraulic efficiency and safety, we performed a multiple linear regression with the same mixed model approach, using hydraulic features as the response variable and the first two PCA axes (PC1 and PC2) and forest type as predictor variables. Data of Ks were log-transformed prior to analysis to achieve normality. More specifically, we expect that the diameter and density of vessels associated with the total intervessel pit membrane area determine transport efficiency, and these xylem features associated with vessel wall thickness, parenchyma, and fiber area determine the variation in hydraulic safety across liana species.

We validated the models by visually verifying the homogeneity of variance and normality of the residuals. For all statistical analyses, we used R v.3.3.0 (R Core Team 2020) with base packages and the lmer() function of lme4 [91].

## Figures and Tables

**Figure 1 plants-14-02951-f001:**
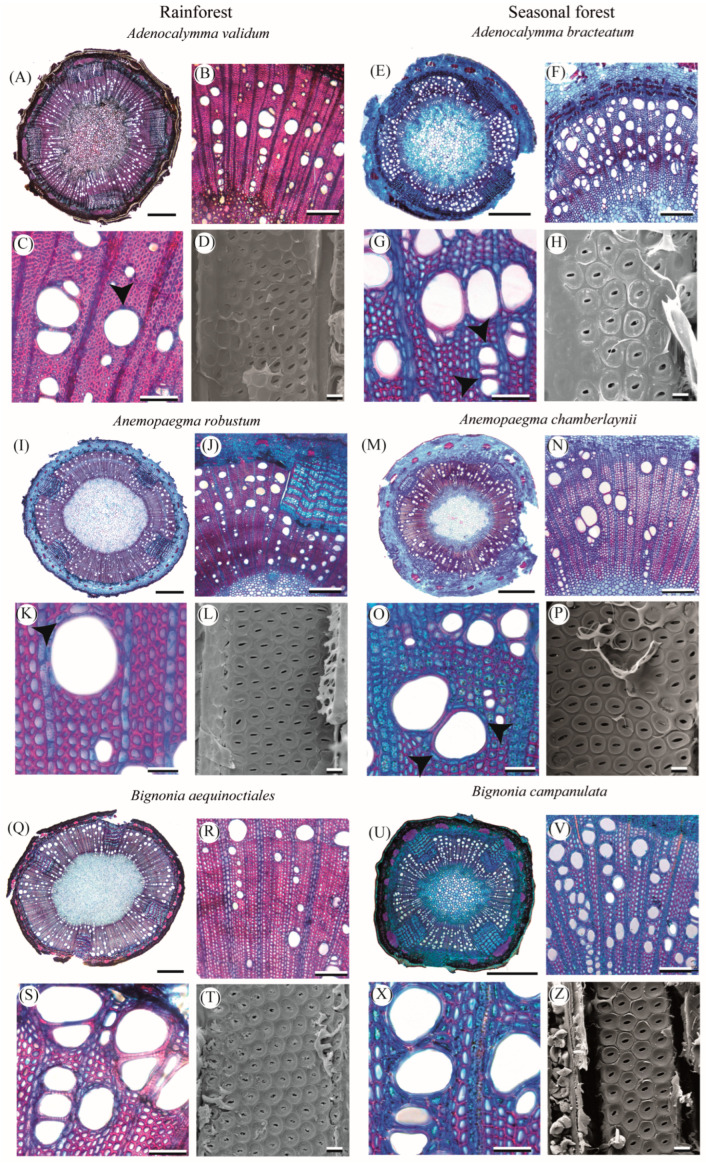
Comparison of the anatomical architecture of young branches of three pairs of congeneric Bignonieae lianas from the rainforest and the seasonal dry forest. (**A**,**E**,**I**,**M**,**Q**,**U**) show transverse sections of the stem with four phloem wedges furrowing the secondary xylem in all liana species (1 mm scale bar; see Appendix A for more details). In (**B**,**C**), (**F**,**G**), (**J**,**K**), (**N**,**O**), (**R**,**S**), (**V**,**X**), the anatomy of secondary xylem (200 µm–50 µm scale bars, respectively). In (**D**,**H**,**L**,**P**,**T**,**Z**), intervessel pits are bordered and alternate, viewed in a longitudinal section of a vessel with a scanning electron microscope (5 µm scale bar). Arrowheads in (**C**,**K**) indicate scanty paratracheal axial parenchyma, and in (**G**,**O**) vasicentric axial parenchyma.

**Figure 2 plants-14-02951-f002:**
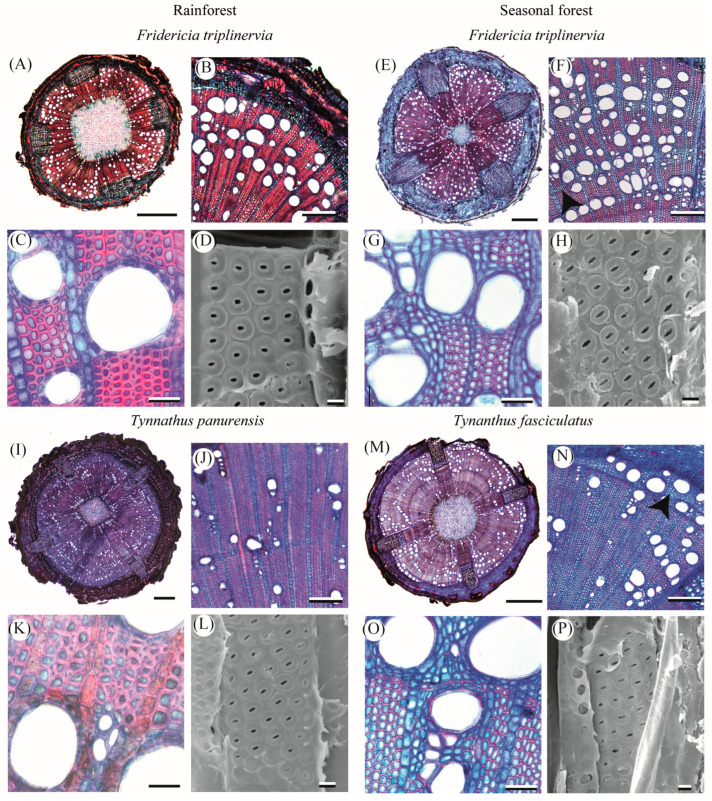
Comparison of the anatomical architecture of young branches of two pairs of congeneric Bignonieae lianas from the rainforest and the seasonal dry forest. In (**A**,**E**,**I**,**M**), transverse section of the stem with four phloem wedges furrowing the secondary xylem in all liana species (1 mm scale bar). In (**B**,**C**), (**F**,**G**), (**J**,**K**), (**N**,**O**), the anatomy of secondary xylem (200 µm–50 µm scale bars, respectively). In (**D**,**H**,**L**,**P**), intervessel pits are bordered and alternate, observed in a longitudinal section of a vessel with a scanning electron microscope (5 µm scale bar). Arrowheads in (**F**,**N**) indicate marginal parenchyma bands.

**Figure 3 plants-14-02951-f003:**
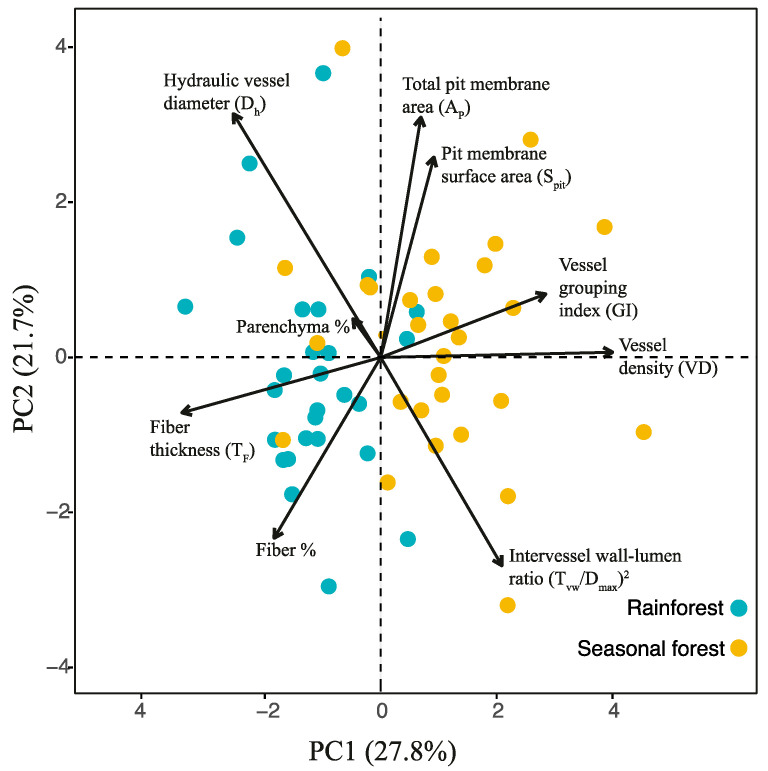
Principal component analysis (PCA) of xylem features of the stems of five pairs of congeneric Bignonieae liana species distributed in two distinct forest types. Biplot with the proportion of covariance recovered by each principal component is described in parentheses in each axis title. The length of the vectors (dark lines radiating from the center) indicates the strength of the influence of each variable on the overall variance of the sample set. More details about the PCA are available in Appendix A.

**Figure 4 plants-14-02951-f004:**
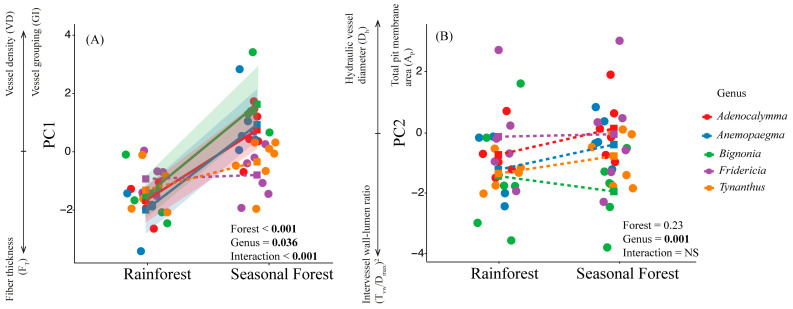
Comparisons of the xylem features, summarized by PCA, across forest types. (**A**) Effect of forest type, genus, and their interaction on xylem features summarized by the PC1 axis. (**B**) Absence of the effect of forest type on xylem features summarized by the PC2 axis. Solid and dotted colored lines, respectively, show significant and non-significant changes between forest types for each genus, based on multiple t-tests with Bonferroni correction (*p* < 0.01; Appendix A) (N = 57). The arrows under each PCA axis indicate the direction of increase in each feature significantly represented by the PCA axes. NS = not significant.

**Figure 5 plants-14-02951-f005:**
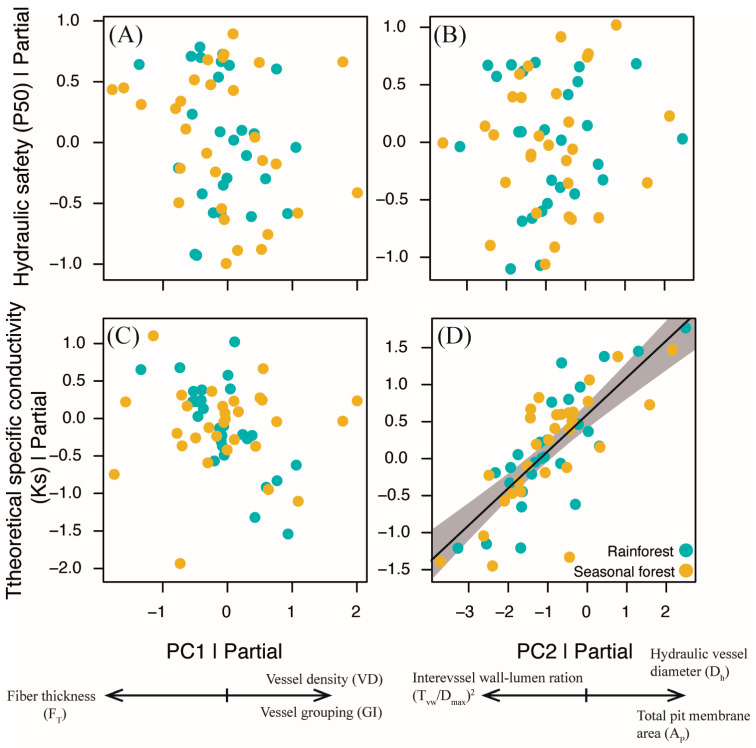
Relationship between hydraulic features and the first two principal components (PC1 and PC2). (**A**,**B**) indicate the absence of a relationship between hydraulic safety (P50) and the PC1 and PC2 axes, respectively. (**C**) Indicates the absence of a relationship between theoretical specific conductivity (Ks) and the PC1 axis. (**D**) Positive relationship between theoretical specific conductivity and the PC2 axis. Additional details of the statistical model applied in each case are available in the results and Appendix A. The arrows under each PCA axis indicate the direction of increase in each feature significantly represented by the PCA axes.

## Data Availability

The data presented in this study are available upon request from the corresponding author.

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
