# Peer review of "Stem Xylem Differences in Congeneric Lianas Between Forests Are Unrelated to Hydraulic Safety but Partly Explain Efficiency"

_plants, 2025, doi:10.3390/plants14192951_

Round 1

Reviewer 1 Report

Comments and Suggestions for Authors

The authors prepared the manuscript on topic Stem Xylem Differences in Congeneric Lianas Between Forests Are Unrelated to Hydraulic Safety and Efficiency. The purpose of the manuscript is not entirely clear, hypotheses are raised, but a clear goal is missing so that the reader immediately understands what question is being addressed in the results.

  • In the abstract, clearly state the purpose of the paper. 1-2 introductory sentences about the topic, then a clear purpose for the manuscript.
  • remove lines 108-110
  • The manuscript provides very general discussions, I would suggest considering adding the supplementary material directly to the manuscript, it provides more information than the data currently in the manuscript. The supplementary material has clearer figures, with clear explanations, and tables with specific data.
  • In the discussion section (mostly in the first paragraph, lines 218 -228), you mention "our results" a lot, but there is no reference to specific results. Please point out the figures and tables where that data is presented.
  • I also recommend separating the conclusions section and briefly presenting the main conclusions. Currently, the general discussion |(I wouldn't call it conclusions) is presented only at the end of the discussion (lines 309-318)

Author Response

Comments 1: The authors prepared the manuscript on topic Stem Xylem Differences in Congeneric Lianas Between Forests Are Unrelated to Hydraulic Safety and Efficiency. The purpose of the manuscript is not entirely clear, hypotheses are raised, but a clear goal is missing so that the reader immediately understands what question is being addressed in the results.

Response 1: Thanks for the suggestion. We've improved the abstract and the last paragraph of the introduction, making the purpose of the study clearer (lines 20-23 and lines 97-99).

Comments 2: In the abstract, clearly state the purpose of the paper. 1-2 introductory sentences about the topic, then a clear purpose for the manuscript.

Response 2: Thanks for the suggestion. We've improved the abstract and the last paragraph of the introduction, making the purpose of the study clearer (lines 20-23 and lines 97-99).

Comments 3: remove lines 108-110

Response 3: Thanks for the suggestion. Done that.

Comments 4: The manuscript provides very general discussions, I would suggest considering adding the supplementary material directly to the manuscript, it provides more information than the data currently in the manuscript. The supplementary material has clearer figures, with clear explanations, and tables with specific data.

Response 4: We agree with the reviewer and have included the main data table (line 163-164 table 1), which was previously in the supplementary material, in the main text. However, we believe that including some figures would result in redundant information.

Comments 5: In the discussion section (mostly in the first paragraph, lines 218 -228), you mention "our results" a lot, but there is no reference to specific results. Please point out the figures and tables where that data is presented.

Response 5: Thanks for the suggestion. We have inserted this information on line 229, 230 and 234.

Comments 6: I also recommend separating the conclusions section and briefly presenting the main conclusions. Currently, the general discussion |(I wouldn't call it conclusions) is presented only at the end of the discussion (lines 309-318)

Response 6: We separated the conclusions section and improved the text (lines 319-330).

Reviewer 2 Report

Comments and Suggestions for Authors

The hypothesis that xylem anatomy “varies with water availability, which determine hydraulic safety and efficiency” could be phrased more clearly. Please revise to clarify causality and grammatical correctness.

The text says “The vessel diameter is a the central feature in an apparent trade-off.” The phrase “a the central feature” is a typo and should be corrected.

(Lines 70–79):The explanation of the pit area hypothesis is dense and could be better structured. Consider splitting into two sentences to separate the hypothesis itself from the evidence limitations.

The figure captions are very detailed but somewhat repetitive. Consider shortening while moving methodological descriptions (e.g., scale bar specifics) to supplementary material.

The explanation of PC1 and PC2 could be clearer by reporting which traits load most strongly, rather than saying “marginally significant contribution.” Please include exact loading values or p-values in the text for transparency.

(Lines 203–208):The authors state no relationship was found with hydraulic safety, but a positive relationship with hydraulic efficiency. Please provide R² values along with β and t-statistics for completeness.

(Lines 270–276):The text says “our results do not support the pit area hypothesis.” Consider elaborating why your results differ from previous studies—was it due to sample size, forest type, or methodological differences?

(Lines 433–441):When explaining Ks calculations, please clarify whether vessel density and diameter were measured per individual or averaged across individuals, since this affects interpretation of variability.

The introduction mainly cites classical works (Tyree, Carlquist, etc.). Please add recent studies (2022–2024) that discuss liana hydraulics under climate change scenarios to highlight novelty.

The efficiency mechanisms are well-discussed, the ecological implications of the lack of correlation with safety are underdeveloped. Please discuss what this means for liana survival in increasingly drought-prone ecosystems.

Linear mixed models are appropriate, but please clarify how you tested for phylogenetic independence beyond using congeneric pairs. Were any phylogenetic signals tested statistically?

Comments on the Quality of English Language

Overall, the manuscript is clearly written, but there are occasional grammatical errors and awkward phrasings (e.g., “a the central feature,” “concomitantly with a lower ratio”). A thorough language edit by a native English-speaking colleague or professional service is recommended.

Author Response

Comments 1: The hypothesis that xylem anatomy “varies with water availability, which determine hydraulic safety and efficiency” could be phrased more clearly. Please revise to clarify causality and grammatical correctness.

Response 1: Thanks for the suggestion. We've improved the hypothesis in abstract and in main text (lines 23-26 and lines 107-111).

Comments 2: The text says “The vessel diameter is a the central feature in an apparent trade-off.” The phrase “a the central feature” is a typo and should be corrected.

Response 2: Thanks, done that (line 60)

Comments 3: (Lines 70–79): The explanation of the pit area hypothesis is dense and could be better structured. Consider splitting into two sentences to separate the hypothesis itself from the evidence limitations.

Response 3: Done that (lines 71-73)

Comments 4: The figure captions are very detailed but somewhat repetitive. Consider shortening while moving methodological descriptions (e.g., scale bar specifics) to supplementary material.

Response 4: We shortened the figure captions (lines 142-149 and 151-157).

Comments 5: The explanation of PC1 and PC2 could be clearer by reporting which traits load most strongly, rather than saying “marginally significant contribution.” Please include exact loading values or p-values in the text for transparency.

Response 5: We improved the explanation of which variables were found to be significantly important for PC1 and PC2 in the M&M section (lines 470-472) and placed the loadings and p-values ​​in the results section (lines 171-172 and lines 177-178).

Comments 6: (Lines 203–208): The authors state no relationship was found with hydraulic safety, but a positive relationship with hydraulic efficiency. Please provide R² values along with β and t-statistics for completeness.

Response 6: Done that. Because we used a mixed model, both marginal and conditional R² values can be obtained. Here, we present the marginal R² (lines 208 and 211), which represents the proportion of variance explained by the fixed effects only, to highlight the contribution of the model’s predictors. Both R² values, along with additional model details, are provided in Table S4 of the Supplementary Material.

Comments 7: (Lines 270–276): The text says “our results do not support the pit area hypothesis.” Consider elaborating why your results differ from previous studies—was it due to sample size, forest type, or methodological differences?

Response 7: We thank the reviewer for this suggestion and have revised the text to improve clarity. We provide at least two possible explanations for why pit membrane area (pit area hypothesis) and other anatomical traits do not show a relationship with hydraulic safety (lines 286-295).

Comments 8: (Lines 433–441): When explaining Ks calculations, please clarify whether vessel density and diameter were measured per individual or averaged across individuals, since this affects interpretation of variability.

Response 8: Thanks for the suggestion. As mentioned, vessel diameter and density were calculated for each individual. We have revised the text accordingly to clarify this point (Line 458).

Comments 9: The introduction mainly cites classical works (Tyree, Carlquist, etc.). Please add recent studies (2022–2024) that discuss liana hydraulics under climate change scenarios to highlight novelty.

Response 9: Thank you for your comment. We have added four recent references in the Introduction that specifically address liana hydraulics under climate change scenarios:

  • Gerolamo, C.S.; Pereira, L.; Costa, F.R.C.; Jansen, S.; Angyalossy, V.; Nogueira, A. (2024). Lianas in tropical dry seasonal forests have a high hydraulic efficiency but not always a higher embolism resistance than lianas in rainforests. Annals of Botany, 134, 337–350.
  • Medina-Vega, J.A.; Bongers, F.; Poorter, L.; Schnitzer, S.A.; Sterck, F.J. (2021). Lianas have more acquisitive traits than trees in a dry but not in a wet forest. Journal of Ecology, 109, 2367–2384.
  • Smith-Martin, C.M.; Jansen, S.; Brodribb, T.J.; Medina-Vega, J.A.; Lucani, C.; Huppenberger, A.; Powers, J.S. (2022). Lianas and trees from a seasonally dry and a wet tropical forest did not differ in embolism resistance but did differ in xylem anatomical traits in the dry forest. Frontiers in Forests and Global Change, 5, 34.
  • Zhang, K.-Y.; Yang, D.; Zhang, Y.-B.; Liu, Q.; Wang, Y.-S.; Ke, Y.; Xiao, Y.; Wang, Q.; Dossa, G. G. O.; Schnitzer, S. A.; Zhang, J.-L. (2023). Vessel dimorphism and wood traits in lianas and trees among three contrasting environments. American Journal of Botany, 110(4), e16154

Comments 10: The efficiency mechanisms are well-discussed, the ecological implications of the lack of correlation with safety are underdeveloped. Please discuss what this means for liana survival in increasingly drought-prone ecosystems.

Response 10: We thank the reviewer for this suggestion and have revised the text to improve clarity. We provide at least two possible explanations for why pit membrane area and other anatomical traits do not show a relationship with hydraulic safety and discuss what this means for liana survival in increasingly drought-prone ecosystems (lines 286-295).

Comments 11: Linear mixed models are appropriate, but please clarify how you tested for phylogenetic independence beyond using congeneric pairs. Were any phylogenetic signals tested statistically?

Response 11: We thank the reviewer for this insightful comment. We did not perform specific tests of phylogenetic signal (e.g., Pagel's λ) on the anatomical and physiological traits. Instead, we explicitly accounted for phylogeny by designing our study around independent evolutionary replicates and by incorporating genus as a fixed effect in our linear mixed models. Our sampling design was specifically structured for this purpose. We selected congeneric pairs, with one species from a rainforest and its closely related counterpart from a seasonally dry forest. The known biogeographic history of the group indicates that the colonization of dry forests was a more recent event. This design allows us to investigate patterns of anatomical and functional differentiation by treating each genus as an independent evolutionary replicate of the transition into a drier habitat. This method is robust when working with lineages that have a history of multiple independent invasions into new environments, as is the case here. We agree that formal phylogenetic comparative methods are most powerful and appropriate when sampling a broad range of species across the entire clade to understand how traits correlate with evolutionary time. However, that was not the case of our study. Our goal was not to model trait evolution across the entire Bignonieae tribe, for which we lack sufficient sampling. Instead, our focused hypothesis required sampling specific, independent pairs from different genera to test for consistent changes associated with forest type, controlling for shared evolutionary history by comparing within genera. Therefore, while we acknowledge the value of phylogenetic signal tests in a broader context, we are confident that our model specification and replicated paired design represent an appropriate method to address our specific question.

Comments 12: Overall, the manuscript is clearly written, but there are occasional grammatical errors and awkward phrasings (e.g., “a the central feature,” “concomitantly with a lower ratio”). A thorough language edit by a native English-speaking colleague or professional service is recommended.

Response 12: We thank the reviewer for this comment. We have carefully checked the manuscript for grammatical errors and awkward phrasings throughout the text. In addition, we revised the sentences highlighted by the reviewer to improve clarity and fluency, and we also used Grammarly Premium to refine the language quality further.

Reviewer 3 Report

Comments and Suggestions for Authors

Dear Authors

The study compares five pairs of congeneric lianas (Bignonieae) in the Amazon rainforest (RF) and Atlantic seasonally dry forest (SDF), characterizing their xylem anatomy (from the cellular level to the intervessel pits) and relating it to “safety” (P50) and “efficiency” (theoretical Ks). Key findings: SDF shows higher vessel density/aggregation and thinner fibers, while no anatomical trait predicts P50; efficiency (Ks) increases with hydraulic diameter, total pit membrane area, and lower wall-to-lumen ratio, regardless of forest type.

The title seems misleading as it states that xylem differences "are unrelated to safety and efficiency," but the manuscript demonstrates an association with efficiency (Ks) along PC2 (hydraulic diameter ↑, pit area ↑, wall-to-lumen ratio ↓), although independent of forest type. The pattern between RF vs. SDF (PC1) does not explain safety or efficiency, but some traits explain efficiency altogether. It is suggested that the title of the manuscript be reworded.

The efficiency is estimated only theoretically (Hagen–Poiseuille) with Dh and VD; a hydraulic validation (measured Kh/Ks) on a minimal set of samples is missing.

Clarify N per species (N = 57 overall, but 6–8 individuals × 10 species were expected). Explain any missing data/exclusion criteria.

Units and symbols: In Table 1, Dh is in “m,” but diameters and SEM measurements are in µm: align (probable typo). Define ρ_w and ν uniformly in the Ks formula and use consistent notation.

Typos/English: “a the central feature” → “a central feature”; “dimorphism” → “dimorphism”; “benched dehydrated” → “bench-dehydrated”; various minor adjustments in Abstract/Introduction/Methods.

Clarification of Figures 4–5: Consider including rug plots/marginal histograms and 95% CIs on regressions.

Terminology: Use constant “seasonally dry forest (SDF)”/“rainforest (RF)” throughout; check consistency of “Bignonieae” vs. “Bignoniaceae” when referring to tribes vs. families.

Author Response

Comments 1: The title seems misleading as it states that xylem differences "are unrelated to safety and efficiency," but the manuscript demonstrates an association with efficiency (Ks) along PC2 (hydraulic diameter ↑, pit area ↑, wall-to-lumen ratio ↓), although independent of forest type. The pattern between RF vs. SDF (PC1) does not explain safety or efficiency, but some traits explain efficiency altogether. It is suggested that the title of the manuscript be reworded.

Response 1: We thank the reviewer for this important comment. Our intention with the title was to emphasize that the anatomical differences found between forest types (PC1) do not translate into differences in hydraulic safety or efficiency. Indeed, PC2 explained variation in efficiency (through Dh, pit area, and wall-to-lumen ratio), but this axis was unrelated to forest type. Thus, forest-related anatomical differences were not associated with safety or efficiency. We agree that the title may give the impression that xylem traits are entirely unrelated to hydraulic function, and change the title to: “Stem Xylem Differences in Congeneric Lianas Between Forests Are Unrelated to Hydraulic Safety but Partly Explain Efficiency”

Comments 2: The efficiency is estimated only theoretically (Hagen–Poiseuille) with Dh and VD; a hydraulic validation (measured Kh/Ks) on a minimal set of samples is missing.

Response 2:  Unfortunately, we lack experimental measurements of hydraulic conductivity (Ks), and our analysis uses only the theoretical conductivity (Kt) derived from the Hagen–Poiseuille equation, a well-established proxy for maximum potential conductivity. Therefore, we cannot perform the validation requested by the reviewer. A consistent, positive correlation between Kt and Ks has been demonstrated across diverse species, including lianas (e.g., Tyree & Ewers, 1991; Sperry et al., 2005; Alber et al., 2019, Trueba et al. 2019). While Kt systematically overestimates Ks, commonly by a factor of ~0.5 (Ewers & Fisher, 1991; Sperry et al., 2005; Alber et al., 2019), the relative distribution of Kt values among plants reflects that of measured Ks. Therefore, Kt serves as a proxy for assessing hydraulic potential across our samples, even if absolute values are overestimated. We acknowledge that future validation on our specific plant group is needed to refine this relationship. However, given the established correlation in the literature, we used Kt as a basis for the comparative analyses presented in this study.

References

  • Alber, M.; Schuldt, B.; Karimi, Z.; Gärtner, H.; De Micco, V.; Eilmann, B.; Smiljanić, M.; Ræbild, A.; Hansen, J.K.; Børja, I.; et al. (2019). Xylem hydraulic safety and efficiency of 100 woody species. Tree Physiology, 39, 1–16.
  • Chiu, S.-T.; Ewers, F.W. (1993). The effect of segment length on conductance measurements in lianas. Plant, Cell & Environment, 16, 993–1000.
  • Ewers, F.W.; Fisher, J.B. (1991). Why vines have narrow vessels: Histological trends in Bauhinia. Plant Physiology, 91, 1625–1631.
  • Sperry, J.S.; Hacke, U.G.; Wheeler, J.K. (2005). Comparative analysis of end wall resistivity in xylem conduits. Plant, Cell & Environment, 28, 456–465.
  • Trueba, S.; Delzon, S.; Isnard, S.; Lens, F. (2019). Similar hydraulic efficiency and safety across vesselless angiosperms and vessel-bearing species with scalariform perforation plates. Exp. Bot. 70, 3227–3240
  • Tyree, M.T.; Ewers, F.W. (1991). The hydraulic architecture of trees and other woody plants. New Phytologist, 119, 345–360.

Comments 3: Clarify N per species (N = 57 overall, but 6–8 individuals × 10 species were expected). Explain any missing data/exclusion criteria.

Response 3: We collected a total of 69 individuals for analysis; however, complete data were not available for 12 of them. These individuals were excluded from the PCA, leaving 57 samples. This information has been included in the Materials and Methods section (lines 470–472)

Comments 4: Units and symbols: In Table 1, Dh is in “m,” but diameters and SEM measurements are in µm: align (probable typo). Define ρ_w and ν uniformly in the Ks formula and use consistent notation.

Response 4: Thanks for the suggestion. These corrections are reflected in the tables of the main text and figures of the Supplementary Material.

Comments 5: Typos/English: “a the central feature” → “a central feature”; “dimorphism” → “dimorphism”; “benched dehydrated” → “bench-dehydrated”; various minor adjustments in Abstract/Introduction/Methods.

Response 5: We thank the reviewer for this comment. We have carefully checked the manuscript for grammatical errors and awkward phrasings throughout the text. In addition, we revised the sentences highlighted by the reviewer to improve clarity and fluency, and we also used Grammarly Premium to refine the language quality further.

Comments 6: Clarification of Figures 4–5: Consider including rug plots/marginal histograms and 95% CIs on regressions.

Response 6: Thanks for the suggestion. We've included the 95% confidence interval in Figures 4 and 5.

Comments 7: Terminology: Use constant “seasonally dry forest (SDF)”/“rainforest (RF)” throughout; check consistency of “Bignonieae” vs. “Bignoniaceae” when referring to tribes vs. families.

Response 7: Done that.

Round 2

Reviewer 1 Report

Comments and Suggestions for Authors

The authors have made most of the essential revisions based on the reviewers' comments. I have no further essential comments.

Reviewer 3 Report

Comments and Suggestions for Authors